# RNA Sequencing Analysis and Verification of *Paeonia ostii* ‘Fengdan’ *CuZn Superoxide Dismutase* (*PoSOD*) Genes in Root Development

**DOI:** 10.3390/plants13030421

**Published:** 2024-01-31

**Authors:** Jiange Wang, Yinglong Song, Zheng Wang, Liyun Shi, Shuiyan Yu, Yufeng Xu, Guiqing Wang, Dan He, Liwei Jiang, Wenqian Shang, Songlin He

**Affiliations:** 1Zhengzhou Key Laboratory for Research and Development of Regional Plants, College of Landscape Architecture and Art, Henan Agricultural University, Zhengzhou 450002, China; lucky_jiangew@163.com (J.W.); edward_song1989@163.com (Y.S.); wzhengt@163.com (Z.W.); sisyrin@henau.edu.cn (L.S.); xuyufeng_2011@163.com (Y.X.); guiqingw@hotmail.com (G.W.); dandan990111@163.com (D.H.); 2Shanghai Chen Shan Botanical Garden, Shanghai 201602, China; yushuiyan1982@163.com; 3College of Horticulture, Henan Agricultural University, Zhengzhou 450002, China; jiangliwei93@163.com; 4School of Horticulture Landscape Architecture, Henan Institute of Science and Technology, Xinxiang 453003, China

**Keywords:** *Paeonia ostii* ‘Fengdan’, adventitious root, RNA-seq, *PoSOD*, *PoARRO-1*

## Abstract

Tree peony (*Paeonia suffruticosa*) is a significant medicinal plant. However, the low rooting number is a bottleneck problem in the micropropagation protocols of *P. ostii* ‘Fengdan’. The activity of superoxide dismutase (SOD) is closely related to root development. But research on the *SOD* gene’s impact on rooting is still lacking. In this study, RNA sequencing (RNA-seq) was used to analyze the four crucial stages of root development in *P. ostii* ‘Fengdan’ seedlings, including the early root primordium formation stage (Gmfq), root primordium formation stage (Gmf), root protrusion stage (Gtq), and root outgrowth stage (Gzc). A total of 141.77 GB of data were obtained; 71,718, 29,804, and 24,712 differentially expressed genes (DEGs) were identified in the comparison groups of Gmfq vs. Gmf, Gmf vs. Gtq, and Gtq vs. Gzc, respectively. Among the 20 most highly expressed DEGs in the three comparison groups, only the *CuZnSOD* gene (SUB13202229, *PoSOD*) was found to be significantly expressed in Gtq vs. Gzc. The overexpression of *PoSOD* increased the number of adventitious roots and promoted the activities of peroxidase (POD) and SOD in *P. ostii* ‘Fengdan’. The gene *ADVENTITIOUS ROOTING RELATED OXYGENASE1* (*PoARRO-1*), which is closely associated with the development of adventitious roots, was also significantly upregulated in overexpressing *PoSOD* plants. Furthermore, PoSOD interacted with PoARRO-1 in yeast two-hybrid (Y2H) and biomolecular luminescence complementation (BiFC) assays. In conclusion, PoSOD could interact with PoARRO-1 and enhance the root development of tube plantlets in *P. ostii* ‘Fengdan’. This study will help us to preliminarily understand the molecular mechanism of adventitious root formation and improve the root quality of tree peony and other medicinal plants.

## 1. Introduction

The roots of seed plants are essential for absorbing nutrients and water, including embryonic roots (ER), primary roots (PR), lateral roots (LR), and adventitious roots (AR) [1,2,3]. Root development is regulated by various developmental and environmental factors, such as hormones, signal sensing, and related genes [4,5,6,7,8]. Auxin, as an endogenous hormone, plays a central role in the hormonal regulatory network of root development [4]. The *AUX/IAA*s, *PIN*s, and *ARF*s, which regulate auxin levels, can influence root development [9,10,11]. Oxidases and antioxidant enzymes also play crucial roles in root development. Indicator amino acid oxidation (IAAO) can regulate the level of auxin, maintain low activity in the early stage of root formation, and promote the formation of root primordia [12]. In apples, treatment with auxin increases the activities of peroxidase (POD) and superoxide dismutase (SOD), reduces rooting time, and promotes the formation of AR [13]. SOD activity is closely related to the formation of adventitious root primordia during the cutting process of *Torreya grandis* and *Cinnamomum camphora* when treated with auxin [14,15]. Although many studies have shown that oxidases and antioxidant activity are related to root development, the function of related genes in rooting is not clear.

In recent years, RNA sequencing (RNA-seq) has been used to study the molecular mechanism of rooting and identify candidate rooting-related genes [16]. This approach has been applied in studies on *Nelumbo nucifera* Gaertn., *Oryza sativa*, *Glycine max*, and tree peony (*Paeonia suffruticosa*) [17,18,19,20]. In mung bean (*Vigna radiata*), genes related to cellular redox homeostasis and the oxidative stress response play crucial roles in the formation of adventitious roots. This highlights the significance of antioxidant enzyme-related genes in the development of adventitious roots [21]. In tree peonies, certain genes related to plant hormones were identified through transcriptome analysis, and these genes were found to be key in promoting the formation of root primordia [22,23]. In short, the research has mainly focused on the molecular regulation mechanism of hormones on rooting, but there are few studies on the effects of antioxidant enzyme-related genes on rooting.

Tree peony is an important medicinal and ornamental plant with a large market demand. However, its low rooting number is one of the bottlenecks that limits the establishment of its regeneration system and industrial development [24]. Therefore, addressing the issue of its low rooting number, particularly in the development of adventitious roots, has become a pressing concern in the establishment of an effective tissue culture regeneration system and the industrial development of tree peony [25]. Currently, numerous scholars have directed their attention to this research area and conducted a series of related studies, primarily concentrating on the impacts of hormone levels and endogenous phenolic substances. Auxin promotes the formation and elongation of adventitious root primordia in the vascular bundles of the stem base [26,27]. Furthermore, tree peony stems contain a variety of phenolic substances that inhibit the formation of adventitious roots by suppressing the activity of antioxidant enzymes [28]. However, there are few reports on the key regulatory genes involved in root development. *PoARRO-1* was cloned in *P. ostii* ‘Fengdan’ (GenBank: AIB53819.1), and its specific expression during adventitious root development was identified. It has a positive regulatory effect on root development via a mediating hormone [29,30]. Wang et al. [31] also found that SOD activity played an important role in the root development of tree peony. However, there are still issues with root development in tree peony, and the molecular mechanism of its rooting is not fully understood. Given the significant role of SOD activity in root formation, it is crucial to determine the function of SOD-related genes in the process of rooting. This is essential for understanding the molecular mechanism of adventitious root formation in tree peony.

In this study, RNA-seq was used to investigate the rooting mechanism and identify the key genes involved in the rooting process of *P. ostii* ‘Fengdan’. In addition, we identified the differentially expressed genes (DEGs) of *PoSOD*, which are highly expressed during root outgrowth stage. Therefore, we cloned the *PoSOD* and analyzed its expression patterns and functions in the tube plantlet of *P. ostii* ‘Fengdan’. Our findings show that *PoSOD* could increase the number of adventitious roots and that it interacted with PoARRO-1 to regulate the formation of adventitious roots in *P. ostii* ‘Fengdan’. These results may help to uncover the molecular mechanism of adventitious root differentiation in tree peony. Furthermore, the results of this study can serve as a theoretical foundation for further research on the rooting mechanism of tree peony and the development of a rapid propagation system for the tissue culture of tree peony and other medicinal plants.

## 2. Results

### 2.1. Quality Analysis of RNA-seq of Four Root Developmental Stages in P. ostii ‘Fengdan’

To identify the genes expressed during the root development, we used the roots at four crucial stages, including the early root primordium formation stage (Gmfq), root primordium formation stage (Gmf), root protrusion stage (Gtq), and root outgrowth stage (Gzc), for RNA-seq. Through the DNBSEQ platform, a total of 141.77 Gb data were obtained. After assembly and removing redundancy, 193,561 unigenes were obtained. In total, approximately 72.32–82.77 M clean reads were obtained. Then, the percentages of Q20 and Q30 were used to illustrate the base-calling accuracy. The percentage of Q20 was above 96%, and the percentage of Q30 was above 86% (Appendix A). These results indicate that the quality of the transcriptome sequencing was good.

### 2.2. Annotation of Database

The annotation rates were made as follows: NR (Non-Redundant Protein Sequence Database) and NT (Nucleotide Sequence Database) as 44.17% and 31.28%, respectively; SwissProt as 32.47%; KEGG (Kyoto Encyclopedia of Genes and Genomes), KOG (Pfam and Clusters of Orthologous Groups for Eukaryotic Complete Genomes), and GO (Gene Ontology) as 35.01%, 36.39%, and 24.64%, respectively; and Pfam as 33.70%. In total, 50.44% of the genes were annotated (Appendix A).

An analysis of the species distribution of 85,497 unigenes annotated to the NR protein database for the best matches showed *Vitis vinifera* had the highest similarity (22.39%), followed by *Juglans regia* (4.99%), and *Nelumbo nucifera* (3.21%) (Appendix A).

### 2.3. Identification and Expression of DEGs

To identify the DEGs during the root development of *P. ostii* ‘Fengdan’, three pairwise comparisons were conducted: Gmfq vs. Gmf, Gmf vs. Gtq, and Gtq vs. Gzc. A total of 71,718 DEGs were identified in Gmfq vs. Gmf, with 48,031 upregulated and 23,687 downregulated genes. Similarly, there were 29,804 DEGs in the Gmf vs. Gtq group, with 21,165 upregulated and 8639 downregulated genes. In the Gtq vs. Gzc group, there were 24,712 DEGs, with 13,556 upregulated and 11,156 downregulated genes (Figure 1A, Appendix A).

Furthermore, 7547 genes were identified as being expressed in the three comparison groups through cross-comparisons. Additionally, 46,011, 7624, and 5564 genes were distinctively expressed in the Gmfq vs. Gmf, Gmf vs. Gtq, and Gtq vs. Gzc comparisons, respectively (Figure 1B). The expression trends of the DEGs varied in the three comparison groups (Figure 1C).

### 2.4. GO Classification and KEGG Enrichment of DEGs

GO enrichment analysis was utilized to categorize the functions of DEGs. In the three pairwise comparison groups, DEGs were found to be involved in cellular processes, metabolic processes, and biological regulation within biological processes. They were also associated with cells, cell parts, and membranes in cellular components, as well as catalytic activity, binding, and transporter activity in molecular functions (Figure 2A–C, Appendix A).

KEGG analysis was further conducted to gain insight into the molecular interactions among the identified DEGs. DEGs in the Gmfq vs. Gmf, Gmf vs. Gtq, and Gtq vs. Gzc groups were assigned to 133, 131, and 132 KEGG pathways, respectively. In the Gmfq vs. Gmf group, 24 KEGG pathways were enriched (Q value ≤ 5%), including plant hormone signal transduction (ko04075) and oxidative phosphorylation (ko00190) (Figure 2D, Appendix A). DEGs in the Gmf vs. Gtq group were enriched in 11 pathways, including phenylpropanoid biosynthesis (ko00940), diterpenoid biosynthesis (ko00904), and sphingolipid metabolism (ko00600) (Figure 2E, Appendix A). DEGs in the Gtq vs. Gzc group were significantly enriched in phenylpropanoid biosynthesis (ko00940), nitrogen metabolism (ko00910), and arginine biosynthesis (ko00220) (Figure 2F, Appendix A).

### 2.5. Analysis of Genes Related to Antioxidant Enzymes

Antioxidant enzymes play a crucial role in root development. Under the condition of Fragments Per Kilobase of Exon Per Million Reads Mapped (FPKM) ≥ 1, ten genes, including *SODCC*, *CSC*, and *SODCP*, related to *SOD* were observed. All the *SOD*s were highly expressed in the Gmfq stage. Six genes related to *ascorbate peroxidase* (*APX*) including *APX1*, *APX5*, *APX3*, *and APX6* were screened out, and *APX1* was highly expressed in the four stages of root development. In total, 25 genes, including *PER42*, *PER72*, and *PER64*, related to *POD* were identified, and most of the *POD*s were highly expressed in the Gmfq stage. Five genes related to *glutathione peroxidase* (*GPX*) were screened, all the *GPX*s were highly expressed in the Gmfq stage (Figure 3A). A more sophisticated protein–protein interaction (PPI) network involving 10 key proteins, including *CCS*, *CSD1*, *GPX8*, *APX1*, *GPX3*, *GPX2*, *GPX5*, *APX3*, *APX6*, and *PER21*, was predicted (Figure 3B). *CSD1* was screened as a central node protein, suggesting that the *CuZnSOD* gene occupies a central regulatory position.

To identify DEGs related to antioxidant enzymes, we identified the 20 most highly expressed DEGs in three pairwise comparison groups (Gmfq vs. Gmf, Gmf vs. Gtq, and Gtq vs. Gzc). No DEGs related to antioxidant enzymes were found in the Gmfq vs. Gmf and Gmf vs. Gtq groups (Figure 4A,B). It was evident that *CL2717.Contig2_All* (*SODCC*) was the only highly expressed *CuZnSOD* gene (Figure 4C), suggesting that *CuZnSOD* may have a significant effect on root elongation. These results indicate that *CL2717.Contig2*_*All* may be involved in root development.

### 2.6. The Validation of RNA-Seq Data via Quantitative Reverse Transcription–Polymerase Chain Reaction (qRT–PCR)

To verify the reliability of RNA-seq data, we performed a further analysis of the gene expression patterns of randomly selected genes using qRT-PCR. The results indicate that the expression patterns of these seven DEGs were consistent with the RNA-seq data (Figure 5).

### 2.7. Cloning and Analysis of PoSOD

To investigate the role of CuZnSOD (CL2717.Contig2_All) in the root development of *P. ostii* ‘Fengdan’, eight *PoSOD* sequences were cloned from *P. ostii* ‘Fengdan’, and it was verified that the sequences were consistent with the transcriptome sequence after aligning the amino acid sequences (Appendix A). We named it *PoSOD* based on the phylogenetic analysis (Figure 6A). Amino acid sequence alignment indicated that PoSOD is homologous to VvSOD from *Vitis vinifera* (Figure 6B). The coding sequence (CDS) of PoSOD was 453 bp, the transmembrane domains and signal peptide prediction indicated that the protein did not contain any transmembrane domain or signal peptides (Appendix A). The PoSOD protein contained a PLN02386 domain (Figure 6C). CuZnSOD from *Lycopersicon esculentum* was used as a template for homology modeling. The sequence identity between PoSOD and the template was 66.23%. The Global Model Quality Estimation (GMQE) was 0.89, and the Qualitative Model Energy Analysis (QMEAN) was 0.86. The sequence consistency with the template was over 30%, indicating that the three-dimensional structure model of PoSOD protein was more accurate (Figure 6D). The expression of *PoSOD* was highest in Gzc (Figure 6E). Furthermore, it was significantly upregulated after treatment with IAA and IBA for 5 d (Figure 6F), suggesting that *PoSOD* may be regulated by auxin.

### 2.8. PoSOD Increased the AR Number in P. ostii ‘Fengdan’ Tube Plantlets

Due to the low rooting rate of tube plantlets, we analyzed the rooting rates of different substrates (Appendix A). Among the substrates, perlite (T2) exhibited the highest level of aeration, while vermiculite (T1) demonstrated the best water-holding capacity (Appendix A). At 45 d of culture, the rooting rate reached 43.3% under the T1 treatment, while the root number of T3 was the highest after 45 days of rooting treatment (Appendix A). At 55 days, the rooting rate and average number of roots for all treatments were higher than those at 45 days; T1 had the highest rooting rate at 60%, and the average number of roots was 2.6 (Appendix A). These results demonstrate that T1 treatment can effectively improve the average number of roots in tube plantlets.

Furthermore, we utilized the *Agrobacterium*-mediated genetic transformation system to introduce *PoSOD* into tube plantlets with T1 treatment (Figure 7A). Compared to CK and transgenic PCAMBIA1302 plants, the expression of *PoSOD* was significantly higher in atransgenic *PoSOD* tube plantlets (Figure 7B). The overexpression of *PoSOD* led to an increase in the average rooting number and rooting rate of transgenic tube plantlets (Figure 7C,D). The activities of POD and SOD in transgenic *PoSOD* tube plantlets were significantly increased (Figure 7E,F). These results also determine that overexpressed *PoSOD* could increase the number of roots in tube plantlets.

### 2.9. Root Phenotype Analysis of Overexpressing PoSOD in Arabidopsis

To further analyze the function of *PoSOD* in roots, we also obtained T3 (transgenic generation 3) stably transformed homozygous lines of *PoSOD* in *Arabidopsis* (Appendix A). The levels of *PoSOD* expression were higher in the transgenic lines (Appendix A). The root lengths of transgenic plants were longer (Appendix A), and the fresh weights of the transgenic lines were significantly higher than those of the wild type (WT) (Appendix A). Nitroblue tetrazolium (NBT) staining analysis revealed that overexpressing *PoSOD Arabidopsis* exhibited weaker NBT signals in the elongation zone, meristem zone, and root cap compared to those with WT (Appendix A). These results demonstrate that *PoSOD* could enhance the root elongation of *Arabidopsis* through scavenging superoxide ion (O_2_^−^) in the elongation zone, meristem zone, and root cap. The overexpression of *PoSOD* significantly enhanced the development of junction adventitious roots (J-AR) under low-light conditions (Appendix A). After treating the rootless of WT and S1 with 10 mg·L^−1^ IBA, the results showed a significant difference in the number of AR between S1 and WT, indicating that *PoSOD* may have a significant effect on AR (Appendix A).

### 2.10. PoSOD Interacted with PoARRO-1

Compared to CK and the transgenic pCAMBIA1302 *P. ostii* ‘Fengdan’ tube plantlet, the expression of *PoARRO-1* was upregulated after overexpressing *PoSOD*, which is consistent with the expression trend of *PoSOD* (Figure 8A). We further analyzed the expression levels of *AtDAO1*(*AT1G14130*) and *AtDAO2* (*AT1G14120*), which are homologous to *PoARRO-1* (Appendix A). Tn transgenic *PoSOD Arabidopsis*, the expressions of *AtDAO1*, and *AtDAO2* were higher than those of WT (Appendix A). These results indicate that the promotion of AR by *PoSOD* in *Arabidopsis* may be related to *PoARRO-1*.

The relationship between PoSOD and PoARRO-1 was verified using a Y2H assay. pBT3-SUC-PoSOD and pPR3-N-PoARRO-1 grew normally on SD/-Leu/-Trp/-His/-Ade medium and were stained blue using X-α-Gal (Figure 8B). Moreover, the yellow fluorescent protein (YFP) signal was observed after co-expressing PoSOD-YFP-N and PoARRO-1-YFP-C through a bimolecular fluorescence complementation (BiFC) assay in *Nicotiana benthamiana* leaves (Figure 8C). Therefore, PoSOD could interact with PoARRO-1 in the cell membranes.

## 3. Discussion

In China, tree peony is known as the king of flowers, and its root bark is a key ingredient in traditional medicine, which is known for its ability to clear heat and remove blood stasis [32]. However, the low propagation coefficient of grafting and cutting significantly impacts its large-scale production. In plants, the increase in root surface area and lateral roots helps absorb nutrients, promoting plant growth and reproduction during root development [33,34]. In this study, we found that DEGs were expressed at four different stages of root development and were enriched in various pathways (Figure 2). During the rooting process, the activities of antioxidant enzymes may enhance the plant’s ability to withstand abiotic stress and contribute to the formation of ARs [35,36,37]. Therefore, researching the root development mechanisms of genes related to antioxidant enzymes will help address the question of the low propagation coefficient. In this study, we showed that several genes encoding antioxidant enzymes are involved in the root development, and *CuZnSOD* can regulate other antioxidant enzyme genes (Figure 3). It is hypothesized that *CuZnSOD* plays a crucial role in regulating root development. A *P. ostii* ‘Fengdan’ *CuZnSOD* was identified from the Gtq vs. Gzc comparison group and named *PoSOD*. It exhibited the highest expression at the Gzc stage (Figure 4C and Figure 6E), suggesting that the gene may play an essential role in root development.

Manganese SODs (MnSODs), CuZnSODs, and iron SODs (FeSODs) protect *Arabidopsis* from the detrimental effects of reactive oxygen species (ROS) [38]. Research has shown that *AtFeSOD1* (*AtFSD1*) increases the number of LRs and positively regulates root development [39]. *AtMnSOD2* (*AtMSD2*) modulates the ROS distribution in *Arabidopsis*, contributing to root skotomorphogenic growth [40]. However, the function of *CuZnSOD* genes has primarily been associated with stress tolerance [12,41]. Furthermore, it is still unclear how *CuZnSOD* affects root growth. In our study, *PoSOD* transgenic *P. ostii* ‘Fengdan’ tube plantlets showed an increase in the number of ARs (Figure 7D). This indicates that *SOD* is necessary for the formation of plant roots, which in *P. ostii* ‘Fengdan’, promotes root development and enhances environmental adaptation. According to a study by Shang et al. [27], 52% of *P. ostii* Fengdan’s tube plantlets successfully rooted under the optimal treatment. In this study, we found that the rooting rate of transgenic *PoSOD P. ostii* ‘Fengdan’ tube plantlets was 56.7%, which did not show a significant difference compared to transgenic pCAMBIA1302 tube plantlets (Figure 7C). After 45 and 55 days of rooting treatment, the rooting rates of the tube plantlets were not significantly different (Appendix A). This suggests that the rooting rate of *P. ostii* ‘Fengdan’ tube plantlets may have reached its upper limit. Researchers can substitute different tree peony varieties to conduct more comprehensive research in the future.

During the rooting process of *Catalpa bignonoides*, increased levels in POD and SOD activity facilitated the formation of AR [42]. Here, the activities of POD and SOD were higher than that of CK after overexpressing *PoSOD* in the tube plantlets (Figure 7E,F). This suggests that *PoSOD* has the potential to enhance the activities of POD and SOD. Research indicates that POD can oxidize endogenous IAA and promote the synthesis of lignin from phenolic substances, facilitating root primordium elongation and root lignification [13,43]. It is speculated that the mechanism of SOD activity may be similar to POD activity in root development. It is suggested that *PoSOD* may promote the increase in adventitious roots by affecting POD activity and subsequently oxidizing IAA. However, more experiments are needed to further verify the results.

Previous studies have reported that antioxidant enzymes may help maintain a stable level by regulating auxin oxidase, but the molecular mechanism remains unknown. Our results show that the overexpression of *PoSOD* increases the expression levels of *PoARRO-1*, which is specifically expressed during adventitious root development, and has a positive regulatory effect on root development in tree peony [29,30]. *AtDAO1* and *AtDAO2*, which are homologous to *ARRO*-1, regulate the oxidation and homeostasis of IAA [44,45,46]. This suggests that *PoSOD* may operate in association with *PoARRO-1*. Here, Y2H and BiFC assays confirmed the interaction between PoSOD and PoARRO-1. This research further confirmed that during the root elongation stage, SOD interacts with ARRO-1 to promote root elongation together (Figure 9). This has been rarely reported in previous studies. However, the regulatory relationship between PoSOD and PoARRO-1 still needs to be verified through further experiments. In summary, we suggest that *PoSOD* may have a similar function to auxin oxidase during root development and may be involved in the auxin oxidation pathway to promote root elongation. However, further research is needed to identify how PoSOD and PoARRO-1 coordinate auxin homeostasis. Our study addressed a knowledge gap regarding the relationship between SOD and auxin oxidation during root development. It also provided a theoretical foundation for investigating the molecular mechanisms of rooting in tree peony and other medicinal plants.

## 4. Materials and Methods

### 4.1. Plant Materials

Materials for RNA-seq were obtained from *P. ostii* ‘Fengdan’ grown in the field at Henan Agricultural University in Zhengzhou, China. Based on morphological and anatomical observations by Sun et al. [30], four crucial stages of adventitious root development were identified. During the root growth period of *P. ostii* ‘Fengdan’ from March to April, four stages were observed (Appendix A). These stages include roots measuring 2–3 cm in length without signs of root primordia bulges (the early stage of root primordium formation/Gmfq), roots longer than 5 cm with no evidence of root primordia bulge (the root primordium formation stage/Gmf), roots with obviously raised root primordia (the root protrusion stage/Gtq), and roots shorter than 2–3 cm (the root outgrowth stage/Gzc). Samples were collected in triplicate at each time point.

The in vitro shoots of *P. ostii* ‘Fengdan’ were transferred to a proliferation medium (MS basic medium with 0.3 mg·L^−1^ 6-benzylaminopurine (6-BA), 0.5 mg/L 1-naphthaleneacetic acid (NAA), 1 g·L^−1^ polyvinyl pyrrolidone (PVP), 50 mg·L^−1^ Vitamin C (Vc), and 0.5 mg·L^−1^ LH) twice, with a subculture cycle of 35 days, under good growth conditions.

WT (Columbia) *Arabidopsis* were preserved by the College of Landscape Architecture and Art at Henan Agricultural University.

### 4.2. RNA Extraction, Library Preparation, Sequencing, Transcriptome Assembly, and Gene Functional Annotation

Total RNAs from samples at four different root growth stages were isolated using the cetyltrimethylammonium bromide (CTAB) method [47]. The total RNAs were verified for their concentration, purity, and integrity through electrophoresis. Then, DNase I was used to remove residual genomic DNA, and the mRNA was purified using magnetic beads with oligo (dT) and then fragmented into small pieces. These RNA fragments were utilized for cDNA synthesis. cDNA libraries were created, and paired-end reads were obtained. We filtered out low-quality reads, those with ambiguous bases, and linker contamination in order to obtain clean reads. Clean raw reads devoid of adaptor sequences and low-quality sequences, including reads containing ambiguous bases (“N”) with more than 50% Q < 10 bases, were assembled using Trinity v2.6.6 software [48]. Next, we assembled the clean reads to obtain unigenes. We conducted a functional annotation and detected SSRs in the unigenes. We also calculated the expression level of each sample based on All-Unigenes [49]. The gene sequences were annotated in various databases, including the KEGG, GO, NR, NT, SwissProt, KOG, and Pfam databases.

### 4.3. Identification and Functional Annotation of DEGs

To identify the DEGs, gene expression levels were analyzed using DESeq and FPKM values. DEGs were selected based on fold-changes ≥ 1 and false discovery rates (FDR) < 0.05. The *p*-value of multiple tests used a false discovery rate (FDR) of 0.05 as the threshold to evaluate the significance of gene expression differences [50].

A GO functional classification analysis was conducted on all DEGs. GO was divided into three functional categories: molecular function, cellular components, and biological processes. KEGG was utilized for pathway enrichment analysis, and Q values ≤ 5% were considered as significant enrichments. Heatmaps were generated using TBtools.

### 4.4. The Prediction of the PPI Network

The DEGs related to antioxidant enzymes that were highly expressed in the proembryo stage were screened to construct a comprehensive primary interaction network. These DEGs were then compared with the STRING v12.0 (https://string-db.org/, (accessed on 4 March 2023)) database to obtain their primary protein–protein interaction network [51].

### 4.5. Isolation of the PoSOD Gene and Bioinformatic Analysis

The *CuZnSOD* gene homolog was isolated from the Gzc cDNA library of *P. ostii* ‘Fengdan’ and designated as *PoSOD*. The CDS sequence was amplified using the primers listed in Appendix A. Amino acid sequences from tree peony and related plant species were aligned using DNAMAN v6.0, and a phylogenetic tree was constructed with MEGA v7.0 using the neighbor-joining (NJ) method with 1000 bootstrap replicates [52]. Conserved domains of PoSOD were predicted using the CDD v3.20 (http://www.ncbi.nlm.nih.gov/Structure/cdd/wrpsb.cgi, (accessed on 15 March 2023)) [53]. Signal peptides were predicted using SignalP v4.1 (http://www.cbs.dtu.dk/services/SignalP/, (accessed on 15 March 2023)) [54], and transmembrane domains were predicted using TMHMM v2.0 (https://services.healthtech.dtu.dk/services/TMHMM-2.0/, (accessed on 17 March 2023)) [55]. The SWISS-MODEL online tool (https://www.swissmodel.expasy.org/, (accessed on 20 March 2023)) was utilized to predict the three-dimensional structure of the protein [56].

### 4.6. P. ostii ‘Fengdan’ Tube Plantlets Treated with Different Substrates

The *P. ostii* ‘Fengdan’ plantlets were transferred into a sterile medium containing 50 mL of rooting mixture, such as vermiculite (T1), perlite (T2), or a 1:1 mixture of vermiculite and perlite (T3). After sterilizing at 121 °C for 30 min, 50 mL of sterile medium (WPM + WPM (Ca^2+^) + PVP (1 g·L^−1^) + IBA (4 mg·L^−1^) + Vc (50 mg·L^−1^)) was added to the sterile matrix (T1, T2, T3). The control consisted of 50 mL sterile medium containing phytagel with 10 tube plantlets per treatment, and the experiment was repeated three times. The average number of roots and the rate of root formation were measured at 45 and 55 days after the start of the rooting treatment.

### 4.7. Transient Expression of PoSOD in P. ostii ‘Fengdan’

The positive monoclonal colonies of pCAMBIA1302 and PoSOD-pCAMBIA1302 were transformed into C58C1 and cultured in a 5 mL LB + Kan (100 mg·L^−1^) + Rif (25 mg·L^−1^) liquid medium at 220 rpm for 2 days. They were then transferred to a 50 mL LB + Kan (100 mg·L^−1^) liquid medium and shaken until the optical density (OD) reached 0.4. The culture was then centrifuged at 5500 rpm for 10 min and resuspended in WPM and 3% sucrose liquid medium. After adding 100 μM acetosyringone (AS) for 2 h, the stem base of the tube plantlets was immersed in the bacterial solution. After being infected for 10 min, the materials were removed and dried on filter paper. The tube plantlets were inoculated in the best rooting medium identified in Section 4.6 and cultured for 2 days. Each treatment involved 10 tube plantlets, and the experiment was repeated three times. After two days of growth in low-light coculture, the infected seedlings were transferred into the optimal rooting medium identified in Section 4.6, which contained Cef (100 mg·L^−1^) + Hyg (5 mg·L^−1^). The medium was replaced every 30 days. The average number of roots and the rooting rate of the tube plantlets under different strains were counted after 45 days of growth. The mixed adventitious roots of six positive transgenic *PoSOD* plantlets were evenly distributed, and the expression of *PoSOD* was determined. The activities of POD and SOD in transgenic plants were determined using Soleibao series kits (BC0090, BC0175, Beijing, China).

### 4.8. Phenotypic Statistics of Roots in Transgenic PoSOD Arabidopsis

The CDS of *PoSOD* was inserted into the *Nco*I/*Spe*I sites of the pCAMBIA1302 vector using the primers listed in Appendix A. The transformation of WT *Arabidopsis* Col-0 was carried out using the floral dip method with GV3101 [57]. Transgenic *Arabidopsis* lines expressing *PoSOD* were identified using qRT-PCR. For further analysis, we utilized T3 homozygous lines that were selected from T2 plants grown on 1/2 MS medium containing 25 mg·L^−1^ hygromycin. The germinated seedlings were then cultivated in a growth chamber at a temperature of 25 °C, with a light cycle of 16 h and a dark cycle of 8 h.

Three generations of transgenic *Arabidopsis* lines (T3) were selected for phenotypic analyses and grown vertically on 1/2 MS medium with 15 replicates for each line. After 7 days, the PR lengths of the transgenic *Arabidopsis* were determined via photography, and the fresh weight was measured. According to the results, the PR lengths of transgenic *Arabidopsis* plants containing pCAMBIA1302 without the *PoSOD* gene were not significantly different from those of WT [58]. Therefore, WT was used as a control. ImageJ version 1.51j8 software was utilized to measure the lengths of the roots. Three biological replicates were conducted for each experiment.

Before sowing on MS medium, the WT and S1 were vernalized at 4 °C for 2 days and then disinfected with 0.1% mercuric chloride for 10 min. The percentage of seedlings with J-ARs was calculated after vertical growth for 10 days under 85 μmol photons m^−2^ s^−1^ light [59]. WT and S1 were grown on 1/2 MS medium for 14 days, and the PRs were removed from the stem base. Subsequently, WT and S1 were transferred to 1/2 MS medium containing 10 mg·L^−1^ IBA for a 3-day dark treatment. Afterward, WT and S1 were transferred back to 1/2 MS medium for 8 days. Each treatment was replicated three times. The number of ARs in the WT and S1 was counted.

### 4.9. NBT Staining

For the determination of superoxide anion radicals, the nitroblue tetrazolium (NBT) staining method proposed by Jambunathan [60] was used. *Arabidopsis* seedlings growing for 7 d were immersed in a 0.5 mg/mL NBT dye solution, and a vacuum pump was used to pump air for 10 min. The NBT dye solution was fully infiltrated into the seedling tissue. The seedlings were dyed in the dark at 28 °C for 2 h. After the seedlings showed blue and purple spots, they were taken out and put into a fixative (ethanol:lactic acid:glycerol = 3:1:1) for decolorization at 90 °C. After the color faded, the seedlings were taken out for observation. The roots were observed directly under a microscope.

### 4.10. Y2H and BiFC Assays

Primers containing *Sif* I sites, as listed in Appendix A, were utilized to amplify PoSOD and PoARRO-1 for the construction of the bait vector pBT3-SUC-PoSOD and the prey vector pPR3-N-PoARRO-1. Subsequently, they were co-transformed into the yeast strain NMY51. Yeast transformants (diluted 10, 100, and 1000 times) were plated on SD/-Leu/-Trp and SD/-Leu/-Trp/-His/-Ade/X-α-Gal media. PoSOD and PoARRO-1 were individually inserted into the *Kpn*I/*Sal*I sites of the pCambia1300-35S::YFPn and pCambia1300-35S::YFPc vectors using the primers listed in Appendix A. For transient expression, GV3101 carrying the pCambia1300-35S::YFPn-PoSOD and pCambia1300-35S::YFPc-PoARRO-1 constructs were simultaneously infiltrated into 5–6-week-old *N. benthamiana* leaves. The interaction of the two proteins in tobacco was observed after 3 days [61].

### 4.11. qRT-PCR

Total RNAs were extracted at different root development stages (Gmfq, Gmf, Gtq, Gzc) from the *P. ostii* ‘Fengdan’, transgenic *PoSOD Arabidopsis*, and *P. ostii* ‘Fengdan’. cDNA was synthesized via reverse transcription using the Evo M-MLV RT Mix kit (AG, Changsha, China). The cDNA was used as the template and quantified via qRT-PCR using the SYBR^®^ Green Pro Taq HS Mix kit (AG, Changsha, China) and the 2^−ΔΔt^ relative quantitative method. The primers utilized in this study were designed using Primer Premier 5 software and are listed in Appendix A.

### 4.12. Data Analysis

SPSS 19.0 software was used for one-way analysis of variance (ANOVA), and Duncan’s test was employed to assess the significant differences in the experimental data (*p* < 0.05).

## 5. Conclusions

In this study, we established a transcriptome database for the four crucial stages of root development in *P. ostii* ‘Fengdan’. *PoSOD* showed a higher expression in Gzc and was cloned from *P. ostii* ‘Fengdan’. The overexpression of *PoSOD* significantly increased the number of ARs and the activities of POD and SOD in *P. ostii* ‘Fengdan’ tube plantlets. And PoSOD interacts with PoARRO-1. In summary, *PoSOD* could enhance POD and SOD activities, regulate the expression of *PoARRO-1*, and promote the root development of *P. ostii* ‘Fengdan’ tissue culture plantlets. Our study extended the transcriptome data related to the development of tree peony roots and verified the interaction between antioxidant enzymes and auxin oxidase during this developmental process. This study also functions as a point of reference for other medicinal plants encountering challenges in root formation.

## Figures and Tables

**Figure 1 plants-13-00421-f001:**
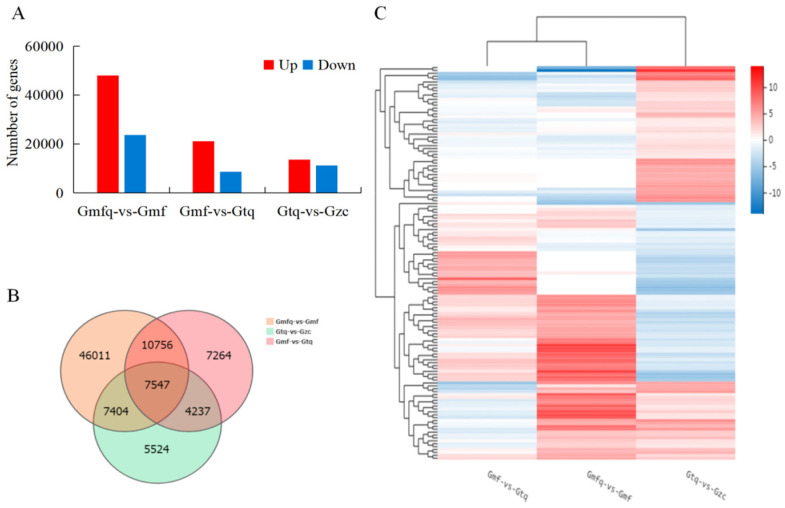
Analyses of DEGs in three pairwise comparisons (Gmfq vs. Gmf, Gmf vs. Gtq, and Gtq vs. Gzc). (**A**) Numbers of upregulated and downregulated DEGs. (**B**) Venn diagram of DEGs. (**C**) Cluster analysis of DEGs.

**Figure 2 plants-13-00421-f002:**
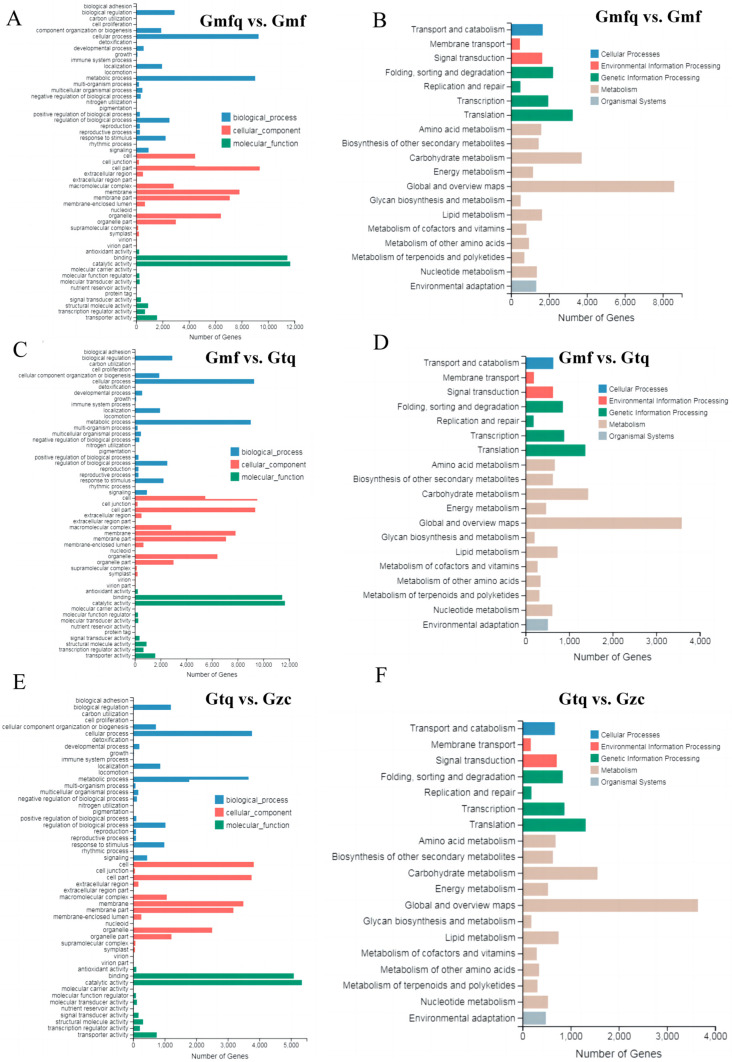
Gene ontology (GO) classification and Kyoto Encyclopedia of Genes and Genomes (KEGG) analysis of DEGs in the three pairwise comparison groups (Gmfq vs. Gmf, Gmf vs. Gtq, and Gtq vs. Gzc). (**A**,**C**,**E**) GO enrichment analyses of three pairwise comparison groups. (**B**,**D**,**F**) KEGG analyses of DEGs in the three pairwise comparison groups.

**Figure 3 plants-13-00421-f003:**
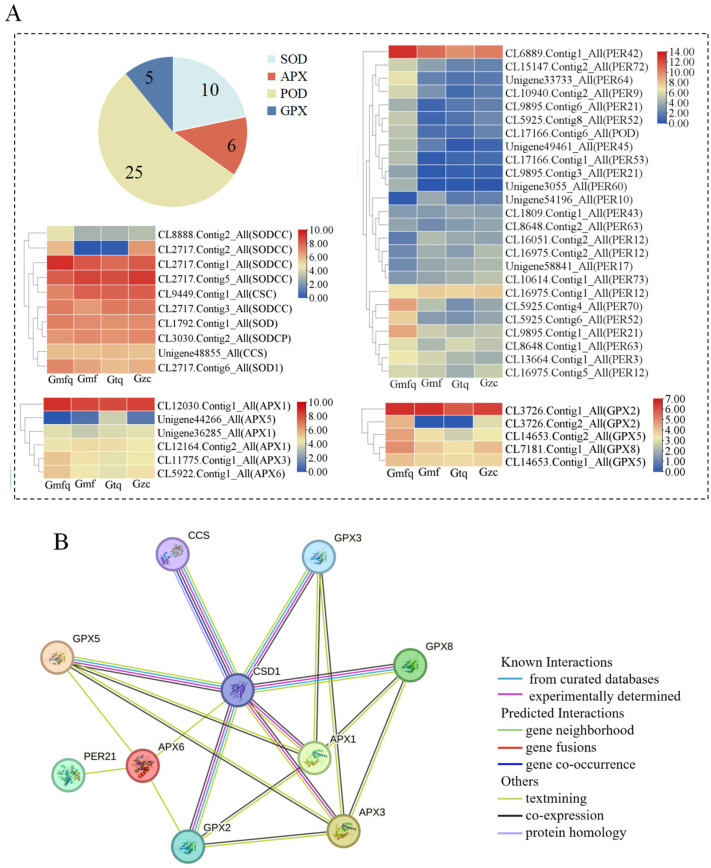
Analysis of genes related to SOD, APX, POD, and GPX during the root development of *P. ostii* ‘Fengdan’. (**A**) Gene number statistics and heatmap analysis. (**B**) The core protein–protein interaction (PPI) network with 10 key proteins in *P. ostii* ‘Fengdan’.

**Figure 4 plants-13-00421-f004:**
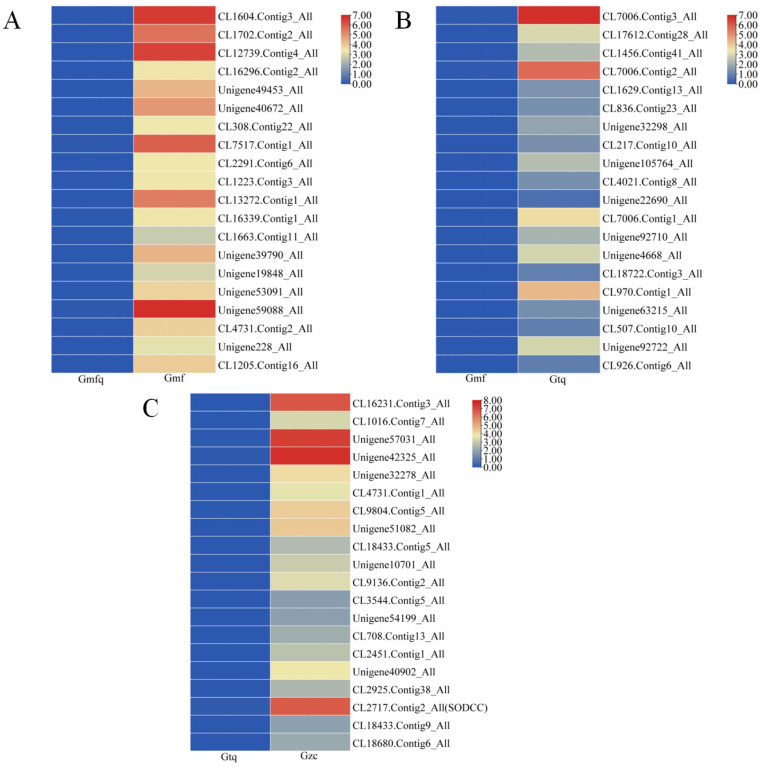
Heatmap of the 20 most highly expressed DEGs in the three comparison groups including (**A**) Gmfq vs. Gmf, (**B**) Gmf vs. Gtq, and (**C**) Gtq vs. Gzc, respectively.

**Figure 5 plants-13-00421-f005:**
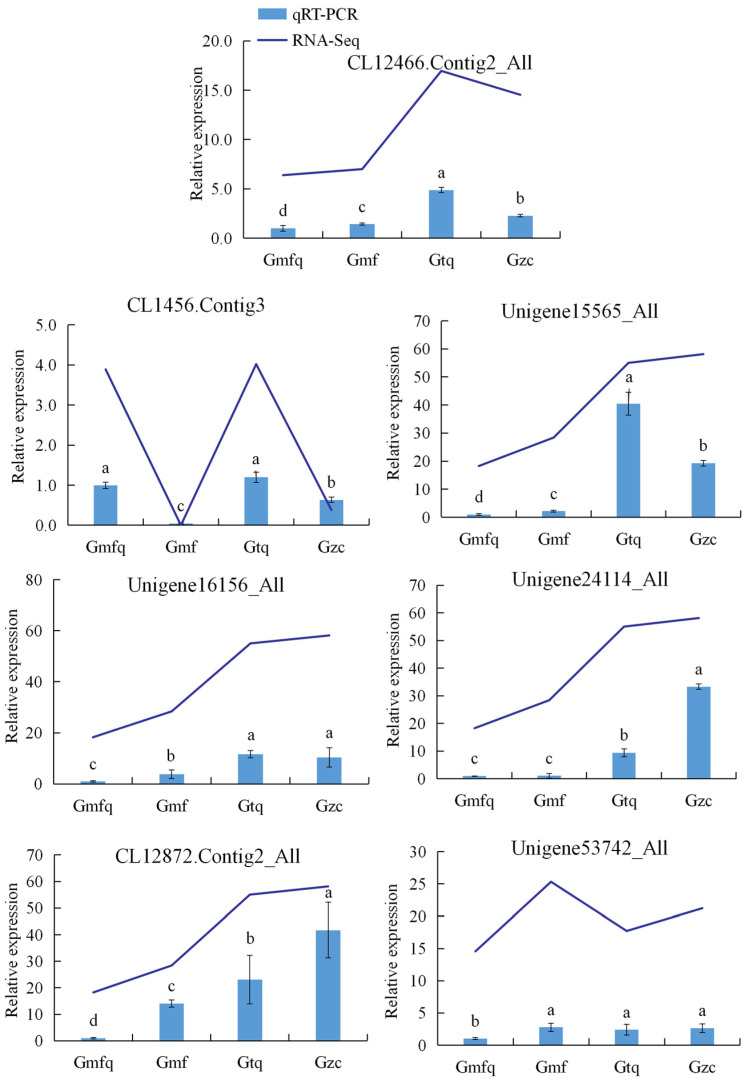
Quantitative reverse transcription–polymerase chain reaction (qRT–PCR) validation of genes selected randomly in RNA-seq results at four different root development stages (Gmfq, Gmf, Gtq, and Gzc). Blue bars show the relative expression levels by qRT-PCR. Blue lines represent the FPKM of RNA-seq. The data were determined using the 2^−∆∆Ct^ method with *Actin* as the reference gene. Bars represent means ± standard errors (SEs); three independent biological replicates were conducted. Different letters represent significant differences between samples based on Duncan’s test, *p* < 0.05.

**Figure 6 plants-13-00421-f006:**
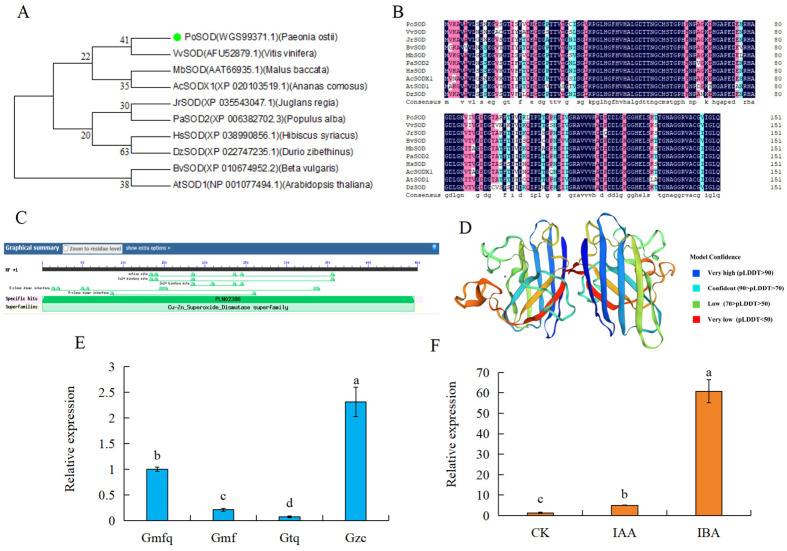
Bioinformatic and expression pattern analyses of PoSOD. (**A**) Phylogenetic tree of SOD proteins from *Vitis vinifera* (AFU52879.1), *Juglans regia* (XP_035543047.1), *Beta vulgaris* (XP_010674952.2), *Malus baccata* (AAT66935.1), *Populus alba* (XP_006382702.3), *Hibiscus syriacus* (XP_038990856.1), *Ananas comosus* (XP_020103519.1), *Arabidopsis thaliana* (NP_001077494.1), and *Durio zibethinus* (XP_022747235.1). Light green circles represented PoSOD. (**B**) Multiple alignments of SOD amino acid sequences. The residues highlighted in dark blue, pink, and light blue indicates 100%, ≥75%, and ≥50% identities, respectively. (**C**) Predicted conserved domains. Light green-shape represented conservative structural domains. (**D**) Tertiary structure of PoSOD. (**E**,**F**) Relative expressions of *PoSOD* in four root development stages (Gmfq, Gmf, Gtq, and Gzc) and various tube plantlet tissues treated with IAA and IBA for 5 d. *Actin* was used as the reference. Different letters represent significant differences between samples based on Duncan’s test, *p* < 0.05.

**Figure 7 plants-13-00421-f007:**
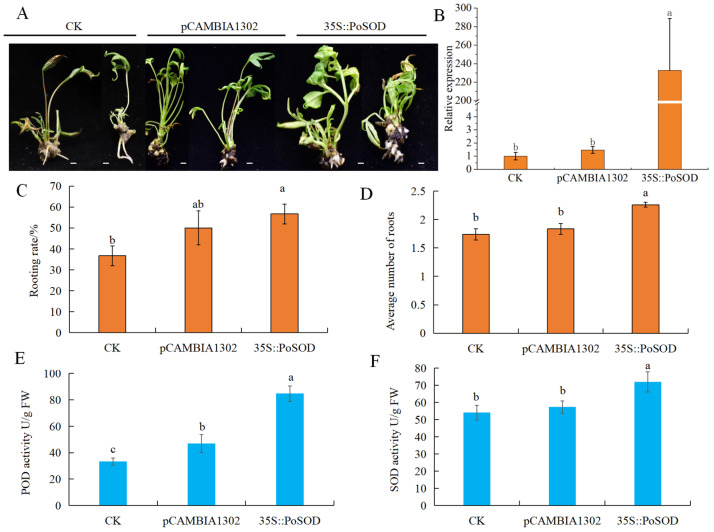
Functional analyses of *PoSOD* in *P. ostii* ‘Fengdan’ tube plantlets. (**A**) Phenotype of CK, transgenic PCAMBIA1302 and *PoSOD* tube plantlet. Relative expressions of *PoSOD* (**B**), average rooting numbers (**C**), rooting rate (**D**), activities of the POD (**E**) and SOD (**F**) of CK, transgenic PCAMBIA1302 and *PoSOD* tube plantlets. Different letters represent significant differences between samples based on Duncan’s test, *p* < 0.05. Scale bar: 1 cm.

**Figure 8 plants-13-00421-f008:**
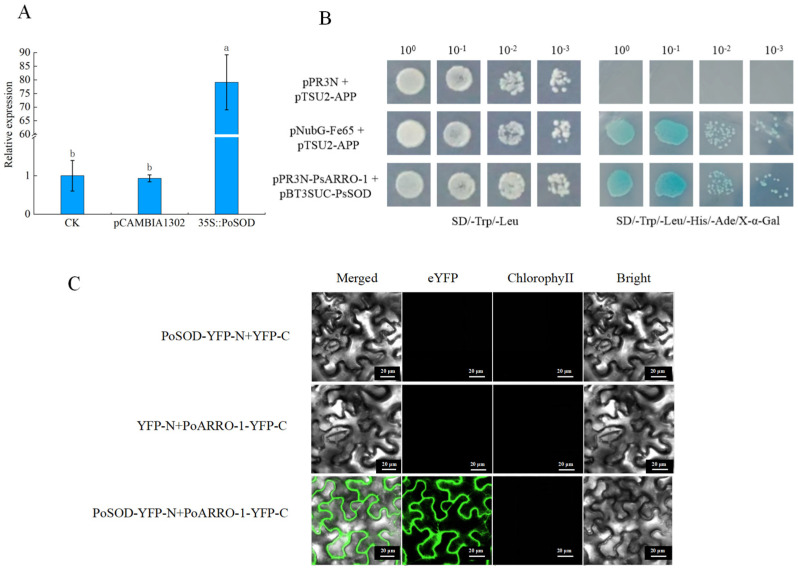
Validation of the interaction between PoSOD and PoARRO-1 (**A**) Relative expression of *PoARRO-1* in control and transgenic *PoSOD* plants. Different letters represent significant differences between samples based on Duncan’s test, *p* < 0.05. (**B**) A Y2H assay showed the interaction between PoSOD and PoARRO-1; 10^0^, 10^−1^, 10^−2^, and 10^−3^ represent yeast transformants diluted 1, 10, 100, and 1000 times, respectively. pNubG-Fe65 + pTSU2-APP was used as the positive control, and pPR3N + pTSU2-APP as the negative control. (**C**) A BiFC assay showed the interaction between PoSOD and PoARRO-1. Scale bar: 20 μm.

**Figure 9 plants-13-00421-f009:**
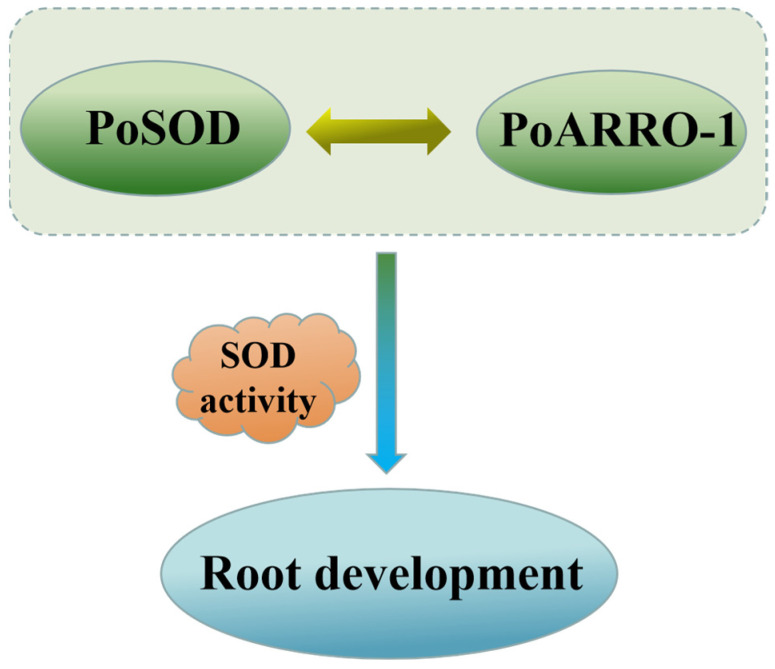
Hypothetical model of the interaction between PoSOD and PoARRO-1 in the regulation of root elongation.

## Data Availability

The data that support the findings of this study are openly available within this manuscript and its Supporting Materials. The raw data of RNA-seq have been uploaded to the NCBI SRA data center (http://www.ncbi.nlm.nih.gov/bioproject, (accessed on 16 November 2023)) under project No. PRJNA1041056.

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
