# Peer review of "RNA Sequencing Analysis and Verification of Paeonia ostii ‘Fengdan’ CuZn Superoxide Dismutase (PoSOD) Genes in Root Development"

_plants, 2024, doi:10.3390/plants13030421_

Round 1

Reviewer 1 Report

Comments and Suggestions for Authors

When gene/protein names are used they font should be carefully checked. All my other comments are included in attatched file.

Comments on the Quality of English Language

Minor editing required

Author Response

Response to Reviewer 1 Comments

Thank you very much for your guidance on our manuscript entitled “RNA Sequencing Analysis and Verification of Paeonia ostii ‘Fengdan’ CuZn Superoxide Dismutase (PoSOD) Gene in Root Development” (ID: 2830667), we appreciate the time and effort that the reviewers dedicated to providing feedback on our manuscript and are grateful for the insightful comments on and valuable improvements to our paper. We have incorporated most of the suggestions made by the reviewers. Please see below for a point-by-point response.

Point 1: The figure is missing.

Response 1: Thank you for pointing out this issue. We have added the Figure S1: Similarity distribution blast to NR annotation of the tree peony (line 122).

Point 2: Please check it, but according to my best knowledge STOP codon does not belong to CDS, so in this case its length should be 453 bp.

Response 2: Thank you for pointing out this issue. We have made revisions (line 212).

Point 3: This finding is inconsistent with the results shown in Figure S3C.

Response 3: Thank you for pointing out this issue. We have made revisions (line 235-243).

Point 4: The sentence is not precise as pCAMBIA1302 plants were also after transformation.

Response 4: Thank you for pointing out this issue. We have made revisions (line 246-251).

Point 5: Such treatment is not mentioned in figure S5c and S5D caption.

Response 5: Thank you for giving us fruitful suggestions. We added a picture, the original Figure S5, to Figure S6. We have added the treatment description in Figure S6.

Point 6: A unification of YFPn and YFPc editing between text and figure 8 caption in needed.

Response 6: We are thankful for your constructive suggestions. We have made revisions in  Figure 8 (line 284 and line 290).

Point 7: Do you mean protein or gene as italics is used to write gene names?

Response 7: Thank you for pointing out this issue. We are so sorry that this part was not clear in the original manuscript. We mean gene as italics to to write gene names, We have reviewed the entire paper and made revisions.

Other changes:

  1. Line214, the statements of “Figure S5” were corrected as “Figure S3”.
  2. Line 220,the statements of “higher” were corrected as “highest”.
  3. Line 274,the statements of “PCAMBIA1302” were corrected as “pCAMBIA1302”.
  4. Line 322,the statements of “transgenic” were corrected as “PoSOD transgenic”.
  5. 5. Line 433, the statements of “4mg·L-1” were corrected as “4mg·L-1”.
  6. 6. Line 528, the statements of “S1–S5: SupplementaryFigures; Tables S1–S8: SupplementaryTables. ” were corrected as “S1–S8: Supplementary Figures; Tables S1–S11: Supplementary Tables”.

In all, I found the reviewer’s comments are quite helpful, and I revised my paper point-by-point. Thank you again for your help!

Reviewer 2 Report

Comments and Suggestions for Authors

The manuscript "RNA Sequencing Analysis and Verification of Paeonia ostii ‘Fengdan’ CuZn Superoxide Dismutase (PoSOD) Gene in Root Development", by Wang et al., provides some interesting preliminary transcriptomic data on adventitious root development in tree peony, there are several major scientific and methodological limitations in experimental design, data analysis, validation assays, result interpretation, and overall presentation as highlighted in my detailed comments.

  1. The introduction lacks critical analysis and fails to clearly define the knowledge gaps in understanding the molecular mechanisms regulating adventitious root formation in tree peony.
  2. The experimental design and methodology lack sufficient details and rationale in several areas. For example, the criteria for selecting the 4 root developmental stages for transcriptomic analysis is not justified.
  3. The RNA-seq data analysis lacks robustness. Important quality control steps like removal of adapter sequences or rRNA depletion are not described.
  4. There is no validation of the RNA-seq data by qRT-PCR to demonstrate reliability of the high-throughput sequencing.
  5. The heatmap analysis clubbing all antioxidant enzyme genes together lacks robust scientific rationale. Their expression profiles across various root stages can be very different.
  6. The cloning of PoSOD is not adequately described and appears rushed without sufficient sequence analysis.
  7. The effect of substrate composition on adventitious root formation lacks scientific controls for valid interpretation of results.
  8. The Agrobacterium-mediated overexpression study in tube plantlets has no negative control or vector alone control to demonstrate the specific effect of PoSOD.
  9. The phenotypic and functional analysis of PoSOD overexpression in transgenic Arabidopsis is disconnected from the main objective of the study.
  10. The link between PoSOD and root meristem regulator PoARRO-1 is claimed without demonstrating any corresponding changes in root morphology or meristem cell responses.
  11. The subcellular localization of interaction is incorrectly interpreted.
  12. The discussion lacks in-depth evaluation of major shortcomings, alternate interpretations and inconsistencies in the data. It is largely focused on supporting the unreliable conclusions.
Comments on the Quality of English Language

Minor english editing required.

Author Response

Response to Reviewer 2 Comments

Thank you for your letter and for the comments concerning our manuscript entitled “RNA Sequencing Analysis and Verification of Paeonia ostii ‘Fengdan’ CuZn Superoxide Dismutase (PoSOD) Gene in Root Development” (ID: 2830667). Those comments are all valuable and very helpful for revising and improving our paper, as well as the important guiding significance to our researches. We have studied comments carefully and have made correction which we hope meet with approval. Revised portion are marked in red in the paper. The main corrections in the paper and the responds to the comments are as flowing:

Point 1: The introduction lacks critical analysis and fails to clearly define the knowledge gaps in understanding the molecular mechanisms regulating adventitious root formation in tree peony.

Response 1: We are thankful for your comments and suggestions. We have added the critical analysis and identified knowledge gaps in understanding the molecular mechanisms that regulate adventitious root formation (line 76-80 and line 85-89).

Point 2: The experimental design and methodology lack sufficient details and rationale in several areas. For example, the criteria for selecting the 4 root developmental stages for transcriptomic analysis is not justified.

Response 2: Thank you for giving us fruitful suggestions. Sun et al. (2022) classified the developmental stages of roots into four phases based on morphological and anatomical observations during root growth. However, they did not investigate the molecular mechanisms underlying these developmental stages. Based on this, this study conducted transcriptome sequencing on the four developmental stages of roots to improve understanding of the molecular mechanisms involved in tree peony root development.

Reference:Sun, Y. K.; Shang, W. Q.; Yuan, J.; Wang, Z.; He, S. L.; Song, Y. L.; Wang. J. G. Functional analysis of PsARRO-1 in root development of Paeonia suffruticosa. Horticulturae, 2022, 8, 903.

Point 3: The RNA-seq data analysis lacks robustness. Important quality control steps like removal of adapter sequences or rRNA depletion are not described.

Response 3: We are thankful for your comments and suggestions. We have added the quality control steps for removing adapter sequences (line 394-396). mRNA was purified using magnetic beads with oligo (dT), this method does not deplete rRNA.

Point 4: There is no validation of the RNA-seq data by qRT-PCR to demonstrate reliability of the high-throughput sequencing.

Response 4: Thank you for giving us fruitful suggestions. The validation of the RNA-seq data by qRT-PCR is shown in Figure 5.

Point 5: The heatmap analysis clubbing all antioxidant enzyme genes together lacks robust scientific rationale. Their expression profiles across various root stages can be very different.

Response 5: We are thankful for your constructive suggestions. We screened the transcriptome for SODs, APXs, PODs, and GPXs genes and analyzed their expression profiles, as depicted in Figure 3A.

Point 6: The cloning of PoSOD is not adequately described and appears rushed without sufficient sequence analysis.

Response 6: Thank you for your advice. We cloned eight PoSOD sequences from the roots of P. ostii ‘Fengdan’ and verified that the sequences were consistent with the transcriptome sequence after aligning the amino acid sequences (Figure S2).

Figure S2. The sequence alignment results of the PoSOD transcriptome sequence and sequencing.

Point 7: The effect of substrate composition on adventitious root formation lacks scientific controls for valid interpretation of results.

Response 7: Thank you for giving us fruitful suggestions. In the substrate treatment of tube seedling rooting, each treatment contained 50 ml of substrate mixed with 50 ml of rooting medium, and the control consisted of 50 ml of rooting medium with phytagel. Substrate screening aims to create an optimal rooting environment, and subsequent transgenic experiments on test-tube seedlings were conducted using the same substrate.

Point 8: The Agrobacterium-mediated overexpression study in tube plantlets has no negative control or vector alone control to demonstrate the specific effect of PoSOD.

Response 8: Thank you for giving us fruitful suggestions. In the Agrobacterium-mediated transformation experiment of in tube plantlets, the negative control comprised the uninfected in tube plantlets, and the transformed pCAMBIA1302 in tube plantlets were depicted in Figure 7.

Point 9: The phenotypic and functional analysis of PoSOD overexpression in transgenic Arabidopsis is disconnected from the main objective of the study.

Response 9: We are thankful for your comments and suggestions. PoSOD is transiently expressed in P. ostii ‘Fengdan’ tube plantlets. The overexpression of PoSOD in Arabidopsis aims to further verify the gene's function in root development. Overexpressing PoSOD can promoted the elongation of PRs and the increase of ARs in Arabidopsis, which further confirmed the role of PoSOD in root development, consistent with the main objective of this study.

Point 10: The link between PoSOD and root meristem regulator PoARRO-1 is claimed without demonstrating any corresponding changes in root morphology or meristem cell responses.

Response 10: We are thankful for your constructive suggestions. We have added the nitroblue tetrazolium (NBT) staining experiment, and the experimental results demonstrate the PoSOD may promote root elongation of transgenic Arabidopsis through scavenging superoxide ion (O2) in the elongation zone, meristem zone, and root cap (Figure S5E).

Figure S5. Phenotypic analysis of PRs in transgenic PoSOD Arabidopsis. (A) Phenotype of seedlings grown in 1/2 MS medium for 7 d after germination, scale bar: 1cm. (B) Relative expression of PoSOD. (C) Root length. (D) Fresh weight. (E) Nitroblue tetrazolium (NBT) staining, scale bar: 100 μM. Different letters represent significant differences between samples by Duncan’s test, P < 0.05.

Point 11: The subcellular localization of interaction is incorrectly interpreted.

Response 11: Thank you for pointing out this issue. We have made revisions (line 287). 

Point 12: The discussion lacks in-depth evaluation of major shortcomings, alternate interpretations and inconsistencies in the data. It is largely focused on supporting the unreliable conclusions.

Response 12: We are thankful for your constructive suggestions. We have added the discussion in the paper.

We tried our best to improve the manuscript and made some changes in the manuscript. These changes will not influence the content and framework of the paper. And here we did not list the changes but marked in red in revised paper.

We appreciate for your warm work earnestly, and hope that the correction will meet with approval.
